# Long-Term Anxiety-like Behavior and Microbiota Changes Induced in Mice by Sublethal Doses of Acute Sarin Surrogate Exposure

**DOI:** 10.3390/biomedicines10051167

**Published:** 2022-05-18

**Authors:** Sabine François, Stanislas Mondot, Quentin Gerard, Rosalie Bel, Julie Knoertzer, Asma Berriche, Sophie Cavallero, Rachid Baati, Cyrille Orset, Gregory Dal Bo, Karine Thibault

**Affiliations:** 1Department of Radiation Biological Effects, Armed Forces Biomedical Research Institute, 91220 Bretigny sur Orge, France; sfm.francois@gmail.com (S.F.); sophie.cavallero@def.gouv.fr (S.C.); 2Micalis Institute, AgroParisTech, Université Paris-Saclay, INRAE, 78350 Jouy-en-Josas, France; stanislas.mondot@inrae.fr; 3Department of Toxicology and Chemical Risks, Armed Forces Biomedical Research Institute, 91220 Bretigny sur Orge, France; gerard@cyceron.fr (Q.G.); rosalie.bel@outlook.com (R.B.); julie.knoertzer@def.gouv.fr (J.K.); asmaberriche@gmail.com (A.B.); 4Institut Blood and Brain@caen-normandie Cyceron, Caen-Normandie University, UNICAEN, INSERM, UMR-S U1237, Physiopathology and Imaging of Neurological Disorders (PhIND), 14000 Caen, France; orset@cyceron.fr; 5CEA, 92260 Fontenay aux Roses, France; 6ICPEES UMR CNRS 7515, Institut de Chimie des Procédés, pour l’Energie, l’Environnement, et la Santé, 67000 Strasbourg, France; rachid.baati@unistra.fr

**Keywords:** 4-nitrophenyl isopropyl methylphosphonate, mood disorder, anxiety, sarin surrogate, sublethal exposure, dysbiosis, microbiota

## Abstract

Anxiety disorder is one of the most reported complications following organophosphorus (OP) nerve agent (NA) exposure. The goal of this study was to characterize the long-term behavioral impact of a single low dose exposure to 4-nitrophenyl isopropyl methylphosphonate (NIMP), a sarin surrogate. We chose two different sublethal doses of NIMP, each corresponding to a fraction of the median lethal dose (one mild and one convulsive), and evaluated behavioral changes over a 6-month period following exposure. Mice exposed to both doses showed anxious behavior which persisted for six-months post-exposure. A longitudinal magnetic resonance imaging examination did not reveal any anatomical changes in the amygdala throughout the 6-month period. While no cholinesterase activity change or neuroinflammation could be observed at the latest timepoint in the amygdala of NIMP-exposed mice, important modifications in white blood cell counts were noted, reflecting a perturbation of the systemic immune system. Furthermore, intestinal inflammation and microbiota changes were observed at 6-months in NIMP-exposed animals regardless of the dose received. This is the first study to identify long-term behavioral impairment, systemic homeostasis disorganization and gut microbiota alterations following OP sublethal exposure. Our findings highlight the importance of long-term care for victims of NA exposure, even in asymptomatic cases.

## 1. Introduction

Several recent high-profile uses of chemical warfare agents (CWA) derived from organophosphorus (OP) compounds have brought them back to attention in the past few years. These include their use in the Syrian conflict (2013–2017) and three separate targeted attacks between 2017 and 2020 resulting in Kim Jong-Nam’s death and the poisonings of Sergei Skripal and Alexei Navalny [1]. These events all took place more than two decades after the first documented uses of OP nerve agents (NA) during the Iran–Iraq armed conflict and in two terrorist attacks in Japan, for which long-term neurological sequelae are still emerging [2,3]. However, the NA dose exposure could not be quantified in either of these reports, and while the described NA exposure was combined with sulfur mustard in Talabani et al. [3], similar symptoms were observed particularly in visual (lacrimation, impaired ability to focus, ocular pain, etc.) and neuropathophysiological areas (headache, numbness, confusion, agitation, etc.) as well as in less-suspected organs (abdominal pain, nausea, and a higher prevalence of respiratory infection). Post-traumatic stress responses were also highly enhanced in sarin-exposed Tokyo victims, correlating with observations made in Gulf War (GW) veterans exposed to high doses of OP (dichlorvos) and carbamate (lindane, bendiocard) pesticides and pyridostigmine bromide (PB) pills and/or NA [4]. These reports therefore highlight the need to meticulously investigate the long-term effects of NA exposure.

Pathophysiologically, NA, OP and carbamate pesticides as well as PB inhibit cholinesterase (ChE), leading to toxic hypercholinergy throughout the body. At the cellular level, acetylcholine accumulation appears at neuromuscular junctions and in synapses, inducing any number of symptoms including fasciculation, hypersecretion, muscle contractions, tetany, tremors and convulsion [5]. Acute exposure to large doses of OP results in muscle paralysis, respiratory distress and ultimately death in only a few minutes post-exposure. This can be avoided by providing an antidote therapy containing a muscarinic cholinergic receptor antagonist (atropine sulfate) and an oxime (pralidoxime, asoxime, etc.) with a strong nucleophile to reactivate the OP-inhibited acetylcholinesterase (AChE). Anticonvulsant medications are also added to the antidote therapy to avoid epileptic seizures and subsequent brain damage [6,7]. However, animal models have revealed that even if the pharmaceutical-based intervention stabilizes the convulsions and improves survival, it is still not enough to avoid long-term neurologic deficits [8]. In addition, other animal models exposed to lower doses of OP have also exhibited long-term neuropsychological dysfunctions, specifically anxiety, depression and cognitive deficits [8,9,10]. These results correlate with the delayed neurological consequences observed in humans after acute OP intoxication, specifically regarding anxiety-related behavior and cognitive deficits [4,11,12,13]. Nevertheless, clarification is needed in animal models, as a single low-dose exposure to OP induces only transient behavioral changes [14] and long-term deficits require concurrent stress [10] or repeated exposure to NA [14,15,16].

Restrictions regulating CWA use and storage are a major restraint to the study of OP-based NA neurotoxicity. Recently, using the sarin surrogate 4-nitrophenyl isopropyl methylphosphonate (NIMP), we developed a murine model to facilitate investigation in CWA-unauthorized laboratories [17]. NIMP is less toxic than sarin, but it has proven itself to be highly effective at inhibiting ChE in vitro and in vivo [18,19,20,21,22]. Furthermore, it can reproduce several features of sarin intoxication, including seizure-like behavior, cortical and hippocampal neuropathologies, neuroinflammatory processes and memory impairment in rodents [17,18,20,22]. These findings thus offer great promise for the study of the consequences of longer-term NIMP exposure in order to better anticipate neurologic sequelae induced by NA exposure.

The aim of this study was to characterize the long-term effects of a single NIMP exposure in mice by evaluating two different sublethal OP doses (0.5 and 0.9 LD50) for over 6 months. For this, we evaluated cerebral and blood ChE inhibition, neuroinflammation, systemic inflammation, behavioral modifications, anatomical and diffusion magnetic resonance imaging (MRI) and gut microbiota impacts at different timepoints for up to 6 months post-exposure. Our results demonstrate that exposure to a low dose of NA has the potential to disrupt gut microbiota and immune homeostasis as well as alter emotional behavior in the long-term.

## 2. Materials and Methods

### 2.1. Animals

All experimental procedures were approved by the SSA animal ethics committee according to applicable French legislation (Directive 2010/63/UE, decret 2013-118). To avoid any potential sexually dimorphic effect, only male Swiss mice (Janvier Labs, Genest-Saint-Isle, France) aged 7 to 8 weeks were housed, with four per cage on a 12 h/12 h light/dark cycle with food and water ad libitum. After a 7-day acclimation period, the animals were randomly assigned to their dedicated experiment. Animal group housing was the same before and after the NIMP challenge. Nine-week-old animals were used for the OP challenge.

### 2.2. NIMP Exposure

4-nitrophenyl isopropyl methylphosphonate (NIMP), used as a sarin surrogate, was synthesized by Dr. Rachid Baati (Université Strasbourg, CNRS UMR 7199, Strasbourg, France) according to a previously reported procedure [21]. On day 0, mice received a single subcutaneous injection (10 mL/kg) of NIMP (LD50 = 0.63 mg/kg), freshly diluted in 0.9% NaCl at two different sublethal doses (0.5 or 0.9 LD50). LD50 was estimated using the improved method of Dixonʹs up-and-down procedure described by Rispin et al. [23]. Control mice (CTL) received a similar vehicle injection. Mouse weight was monitored before the intoxication, every day during the first week following NIMP exposure, twice per week for up to 3 months and then once per week up until 6 months post-intoxication.

### 2.3. Behavioral Observation

#### 2.3.1. Intoxication Severity Scale

All behavioral changes observed in NIMP-exposed mice during the first hour post-intoxication compared to CTL were noted at their onset and considered as observable signs of intoxication. Thirteen intoxication levels (12 signs) were used, grading animals from normal (=0) to death (=12) (Figure 1a). For the entirety of this study, 165 mice were evaluated including 54 CTL mice, 55 mice exposed to 0.5 LD50 and 56 mice exposed to 0.9 LD50 of NIMP.

#### 2.3.2. Anxiety-like Behavior Tests

Before and every month after NIMP exposure, anxiety behaviors were evaluated for each group of mice (CTL *n* = 14; 0.5 LD50 *n* = 15; 0.9 LD50 *n* = 16). Anxiety tests are based on the balance between the natural tendency of mice to explore novel environments and their apprehension for open and bright areas. This approach–avoidance conflict results in behaviors correlated with an increase in physiological stress indicators. A new anxiety test was performed every month to avoid test habituation and exploration diminution during the task. Tests were randomly selected. Due to the similarity of the elevated plus maze (EPM) and elevated zero maze (EZM) tests, these tests were assigned to the first and the last period of the study, respectively.

Open-field test: square area or circular areaMice were placed for 5 min in either an empty square open-field (45 × 45 cm) or circular open-field (40-cm diameter) box surrounded by high walls to prevent escape. The test period was videorecorded, and the activity of the animal over time was analyzed using EthoVision XT software (Noldus, Wageningen, The Netherlands). The distance, speed and time spent walking around the outer edge of the box vs. the center (square or round area depending on the open-field shape) of the box were evaluated.

EPM test or EZM testThe EPM apparatus consists of a raised maze (80 cm off the floor) with four arms in a cross shape: two arms are exposed to the open air, and the other two arms are enclosed. EZM is a circular apparatus with dark enclosed sections alternating with open sections. Mice were placed in the maze for a 5-min period. The test period was videorecorded, and the activity of the animal over time was analyzed using EthoVision XT software. The time spent in the open arms was evaluated.

Staircase testThe staircase contains six identical steps (2.5 cm high and 7.5 cm deep) enclosed between vertical walls (10 cm wide). Wall levels are constant along the staircase. Each mouse was placed individually at the bottom of the staircase for a 5-min observation period. The top step (5th step) is considered to be the most anxious for mice, as it is more elevated and brighter than the others. The number of rearings on the 5th step was recorded and used as the anxiety index.

Dark–light box testThe dark–light box apparatus is divided into two compartments: a light one (white, bright and without a lid) and a dark one (black, closed and covered). Mice were placed in the light compartment and videorecorded for a 5-min period. The time spent in both compartments was evaluated using the EthoVision XT software.

Neophobia testThis test is based on the appetence for sugar in mice and was adapted from the “Novelty Suppressed Feeding Test” without food restriction. Mice were placed in a bright box with chocolate cereal in the center of a platform and videorecorded for a 10-min period. The platform visit frequency was measured using the EthoVision XT software.

### 2.4. Cholinesterase Activity and Multiplex Biomarker Assays

At different timepoints (6 h, 24 h, 3 days, 7 days, 1 month and 6 months) after NIMP exposure, animals were deeply anesthetized with pentobarbital. Immediately afterwards, blood samples (800 µL) were collected by intracardial sampling with a syringe and mice were transcardially perfused with 15 mL of cold NaCl (0.9%). Blood samples were divided and prepared for two separate analyses, with 400 µL containing 60 µL of 1.6% EDTA to avoid platelet aggregation and 400 µL used for serum analyses. Blood samples with EDTA were used to count white blood cells (WBC). Mononuclear and polynuclear cell counts were carried out using an IDEXX ProCyte Dx Hematology analyzer.

Brains were quickly removed and dropped in cold saline buffer before being sliced in 2-mm-thick coronal sections. The piriform cortex and amygdala were dissected, collected in microtubes and frozen in dry ice. Samples were homogenized in 50 mM phosphate buffer (pH 7.4)/0.5% Tween using a bead mill homogenizer (OMNI International) with 1.4-mm ceramic beads and centrifuged at 10,000× *g* (4 °C) for 10 min. The resulting supernatants were stored at −80 °C. Total protein concentrations were determined using the DC Protein Assay (Bio-Rad, Marnes-la-Coquette, France) according to the manufacturer’s protocol.

#### 2.4.1. Cholinesterase Inhibition Assay

Total ChE activity was determined using the Ellman method by adding 5 µL of piriform cortex and amygdala sample to 0.22 mM 5,5′-dithiobis-2-nitrobenzoic acid (DTNB, Sigma Aldrich, St-Quentin Falavier, France) in phosphate buffer (pH 7.4). In parallel, 0.1 mM ethopropazine hydrochloride (Sigma Aldrich) was added for AChE-specific activity analyses. After a 15-min baseline reading to account for thiols present in the samples, 1 mM acetylthiocholine (Sigma Aldrich) was added and the reaction between thiocholine and DTNB was monitored for 30 min at 412 nm and at 25 °C in a microplate reader (Spark 10, Tecan). All samples were assayed in duplicate. Activities of the piriform cortex and amygdala samples were normalized to total protein concentration for each sample. The final results were expressed as percentages of average CTL activity.

#### 2.4.2. Milliplex Multiplex Assays

Cerebral and serum concentrations of KC, IL-1α, IL-10, IL-9, IL-17, GM-CSF, M-CSF and G-CSF were measured using the Milliplex MCYTMAG-70K-PX32 (Merck-Millipore, Burlington, MA, USA) according to the manufacturer’s protocol.

### 2.5. Anatomical Examination

#### 2.5.1. In Vivo MRI and Analysis

MRI scans were performed with a Bruker Biospec 70/30 (7T) preclinical scanner (Bruker Biospin MRI, Ettlingen, Germany). Images were acquired and reconstructed, and parametric maps were generated using Paravision 6.0.1 (Bruker Biospin MRI, Ettlingen, Germany). Mice were imaged at five timepoints (48 h prior to intoxication and 72 h, 7 days, 1 month and 6 months post-NIMP exposure). Immediately prior to imaging, animals were anesthetized with 5% isoflurane and thereafter maintained with 2% isoflurane in a 70%/30% mixture of NO2/O2. Body temperature was maintained at 37 °C, with a respiration rate of 50–70 breaths per min. High resolution T2 sequences were performed using the following parameters: repetition time (TR) = 3500 ms, effective echo time (TE) = 40 ms, field of view (FOV) = 17.92 × 17.92 mm2 (256 × 256 data matrix) and 18 slices with a 0.5-mm thickness. Diffusion-weighted images (DWI) were collected using the following parameters: TR = 2500 ms, TE = 22 ms, FOV = 20 × 20 mm2 (128 × 128 data matrix), eight slices with a 0.8-mm thickness, three diffusion-weighted orthogonal directions with b = 650 s/mm2 and a total acquisition time of 16 min. Apparent diffusion coefficient (ADC) maps were generated and volumes of interest (VOI) were manually traced on ADC parametric maps around the amygdala brain region.

#### 2.5.2. Histology

Six months post-NIMP exposure, mice were deeply anaesthetized with pentobarbital and transcardially perfused with 10 mL of cold NaCl (0.9%), followed by 30 mL of 4% paraformaldehyde (PFA) in phosphate buffer.

Microglia stainingBrains were quickly removed and immersed in cold 4% PFA for overnight post-fixation. Brains were cryoprotected for 24–48 h in cold 20% sucrose and then frozen in −40 °C isopentane. All brains were sliced in 14-µm coronal sections using a cryostat, and slices were sequentially mounted on Superfrost + slides (VWR, Radnor, PA, USA). Immunofluorescent labeling with rabbit anti-IBA1 (1/1000, Wako, USA), detected by anti-rabbit 555 (1/500, Invitrogen, Waltham, MA, USA), was performed on one slide per animal to analyze microglial reactivity. Fluorescent labeling of the amygdala was imaged using an automated Leica DM6000 B research microscope (Leica Microsystems, Wetzlar, Germany), and the two hemispheres of three different slices were analyzed for each animal. All acquisitions were performed using the same acquisition setup. Stereotaxic consistency between animals was maintained with the help of a reference mouse brain atlas [24]. Analyses were performed using the ImageJ software. To quantify Iba1 staining, we performed an area fraction analysis. Briefly, after setting a threshold, the pixels in the image with values inside this range were converted to white, whereas pixels with values outside this range were converted to black. The threshold was determined to obtain a clear area representing IBA1 labeling in CTL animals. The same threshold was applied to each image of all animal groups. This analysis measured the area of labeling in regions of interest.

Gut histologyThe Swiss-rolling technique was used to examine complete colonic sections. This technique helps in the histological assessment of the complete colonic sections examined. The result is an intestinal roll which allows for the scanning of a large part of the intestine. The intestinal villi and the epithelial lining remain intact despite being rolled up. Following PFA fixation, the organs were rinsed with distilled water and dehydrated. Samples were cut at a thickness of 4 µm on a rotary microtome (LEICA^®^). Hematoxylin-eosin-saffron staining was performed on successive sections for structural and functional analysis of the colon. The length of villi, number of goblet cells and area of immune infiltrates were quantified manually using Histolab software (GT Vision, UK).

### 2.6. Gut Microbiota Modification

#### 2.6.1. Gram+/Gram− Ratio Determination

To observe the bacterial microflora, a fecal smear of each animal was performed using the Gram staining technique. For this, the colon was cut lengthwise and the feces were directly removed, diluted in 500 μL of 1X PBS (Gibco) and spread on a slide. Differential staining of Gram+ bacteria from Gram− bacteria was performed using the Gram−Hucker R Kit (RAL Diagnostics). The Gram+/Gram− ratio was determined for the entire smear by microscopy using a 40× objective.

#### 2.6.2. Assessment of Gut Microbiota Composition by High-Throughput Sequencing

DNA extraction, 16S rRNA gene amplification, 16S rDNA amplicon library preparation and sequencing were carried out by SMALTIS (http://www.smaltis.fr/, accessed on 26 September 2019). DNA was extracted from 200 mg of distal colonic luminal content using the QIAamp Fast DNA stool Mini Kit (Qiagen) according to the manufacturer’s recommendations (isolation of DNA from stool for pathogen detection). The V3–V4 region of the 16S rRNA gene was amplified using AccuStart™ II PCR ToughMix (QuantaBio) and the following primers: V3F “CTTTCCCTACACGACGCTCTTCCGATCTACGGRAGGCAGCAG” (344F) and V4R “GGAGTTCAGACGTGTGCTCTTCCGATCTTACCAGGGTATCTAATCCT” (802R). The thermocycler was programmed with an initial DNA denaturing step at 95 °C for 2 min followed by 30 cycles at 95 °C for 1 min, 65 °C for 1 min, 72 °C for 1 min and a final extension step at 72 °C for 10 min. Ligation of MiSeq sequencing adapters was performed by PCR using the MTP™ Taq DNA Polymerase (Sigma) and the following thermocycler conditions: 94 °C for 1 min followed by 12 cycles at 95 °C for 1 min, 65 °C for 1 min, 72 °C for 1 min and a final extension step at 72 °C for 10 min. Sequencing was performed on a MiSeq device using the 2 × 250bp V3 kit. The remaining adapter/primer sequences were trimmed, and reads were checked for quality (≥20) and length (≥200 bp) using cutadapt [25]. Reads were further corrected for known sequencing errors using SPAdes [26] and then merged using PEAR [27]. Operational taxonomic units (OUTs) were identified using a Vsearch pipeline [28] set up to dereplicate (--derep_prefix –minuquesize 2), cluster (--unoise3) and chimera check (uchime3_denovo) the merged reads. OTU taxonomical classification was performed using a classifier from the RDPTools suit [29].

### 2.7. Statistics

#### 2.7.1. General Statistics

Data were expressed as means ± standard deviation (SD) and analyzed using PRISM 7 software (GraphPad, San Diego, CA, USA). Statistical tests and sample sizes are indicated in the figure legends and figures, respectively. Graphs display the mean and error bars represent the SD. The significance between groups is denoted by * *p* < 0.05, ** *p* < 0.01, *** *p* < 0.001 or **** *p* < 0.0001.

#### 2.7.2. Microbiota-Specific Statistics

Statistical analyses were run using the R programming language and software together with the gplots, gdata, vegan (http://cran.r-project.org/package=vegan, accessed on 3 March 2020), ade4, Hmisc, corrplot and phangorn packages. OTU counts were normalized via simple division to their sample size and then multiplied by the size of the smallest sample. α-diversity and richness were estimated using diversity and estimateR. The distance matrix for β-diversity analysis was computed using vegdist and the Bray–Curtis method. Principal coordinates analysis was computed on a distance matrix using dudi.pco. Associations between the microbiota composition at the genus level, anxiety-like behavior measurements and mononuclear cell percentages were assessed using rcorr. The Kruskal–Wallis rank sum test and post hoc Dunn’s all-pairs rank test were used as required to detect differences between groups. *p*-values were adjusted as necessary using false discovery rate correction.

## 3. Results

### 3.1. General Effects of NIMP Intoxication

The severity of NIMP intoxication was evaluated based on our intoxication scale (Figure 1a). This scale includes 12 relevant behavioral changes, from no behavioral perturbation to death, the maximum on our scale. Almost all mice exposed to 0.5 LD50 of NIMP displayed fasciculation and face stereotypies (chewing, yawning), considered as light intoxication symptoms. On the other hand, the majority of the 0.9 LD50-exposed mice presented long-lasting convulsions. This dose was thus selected to be the high sublethal dose in our study. The mean intoxication scores were 2.5 ± 0.2 for 0.5 LD50 and 10.4 ± 0.2 for the 0.9 LD50-exposed mice (Figure 1b). Six animals (10.7%) exposed to the highest dose and none exposed to the lowest dose died in the first day following the NIMP challenge. No delayed death was observed.

As expected, measurements of cerebral ChE activities in mice exposed to 0.9 LD50 revealed a significant important total ChE inhibition 6 h after intoxication (5.4 ± 0.6% of ChE activity). Late measurements showed a persisting significant inhibition until 1-month post-intoxication (72.5 ± 6.7% of ChE activity), with recovery being complete 6 months post-exposure (Figure 1c). Exposure to 0.5 LD50 induced a large inhibition of ChE activity (36.9 ± 4.6%) at 6 h post-exposure. The recovery appeared incomplete at 1 month (81.7 ± 5%) and 6 months after exposure at this dose (77 ± 13%); however, it was not significant from CTL activity (Figure 1c). Interestingly, the profile of AChE activity was the same as the total ChE activity for both doses (Figure 1d), suggesting that in the CNS, the majority of ChE activity is due to AChE activity.

Finally, the evolution in weight gain during the 6 months post-exposure revealed a significant loss of weight for only the 0.9 LD50-exposed mice during the first week after intoxication. One month after exposure, the change in mean body weight of the exposed animals reached the change in body weight of the CTL group. Interestingly, the 0.5 LD50-exposed mice did not display any difference in weight gain at any time compared to the CTL group (Figure 1e).

### 3.2. Anxiety-like Behavior Modification

Anxiety-related behaviors were evaluated every month after NIMP exposure. To avoid any test habituation, a new anxiety test was performed for each measurement. The initial test, which was performed before any intoxication, showed no difference in terms of the time passed exploring the center zone in the square area open-field and was thus considered as the anxiety level baseline (Figure 2a). One month after NIMP intoxication, the time passed in the open arms of the EPM significantly decreased in the NIMP-intoxicated animals compared to the CTL (Figure 2b). The number of rearings on the 5th step of the staircase measured on the second month post-exposure did not show any significant difference between groups, even though this number was slightly decreased in both NIMP intoxication groups (Figure 2c). In contrast, the recovery was complete 3 months post-exposure, since the time passed in the light box during the dark–light box test was equivalent between groups (Figure 2d). Four months post-intoxication, the time passed exploring the center zone in the circular area open-field decreased drastically in the 0.9 LD50-exposed mice group, while the 0.5 LD50-exposed mice did not show any difference with the CTL group (Figure 2e). Finally, the two NIMP-intoxicated animal groups at 5 and 6 months post-NIMP exposure showed a significant increase in the anxiety index, as measured by the decreased frequency of food zone exploration in the neophobia test and the time passed in the open arms in the O-maze (Figure 2f,g).

### 3.3. Longitudinal Imaging Examination and Long-Term Induced Inflammation

To explain these behavioral long-term modifications, we conducted MRI in parallel on 0.5 and 0.9 LD50 NIMP-exposed mice for 6 months to evaluate morphological modifications and brain edema after exposure. The image analysis did not show significant neuroanatomical changes using T2-weighted images or edema formation using ADC mapping on DWI images (Figure 3).

Next, cerebral cytokine levels were evaluated in the amygdala/pyriform cortex extracts using Luminex technology. A slight increase in several cytokine levels was observed in the NIMP-exposed mice group 6 months post-exposure (Figure 4a). Thus, sublethal doses of NIMP did not induce any robust long-term neuroinflammation. In addition, histological exploration showed no persistent microglia reactivity 6 months post-intoxication in the amygdala of exposed animals (Figure 4b,c), confirming the previous cytokine results.

Because the impact of a long-term systemic inflammation could alter the nervous system, we evaluated the evolution of WBC after NIMP exposure. A first phase could be observed, with significant WBC modifications in the 0.9 LD50-exposed mice group. Two timepoints, one day and 7 days after NIMP exposure, showed significant decreases in lymphocytes, monocytes and eosinophils in the highest dose-exposed group, whereas the lowest dose induced little or no modifications in most cell types (Figure 5). One-month post-intoxication, no difference was observed between the NIMP-exposed groups or in comparison to the CTL group. Surprisingly, a second phase was also observed between 3 and 6 months post-NIMP exposure, with significant modifications to WBC quantities in the 0.5 LD50-exposed mice group. With the exception of basophils and eosinophils, which were significantly decreased in the two exposed groups, the 0.5 LD50-exposed mice group presented significant long-term modification in the proportion of lymphocytes, monocytes and neutrophils compared to the 0.9 LD50-exposed and CTL groups (Figure 5).

Although serum cytokine levels were not significantly increased 6 months post-exposure, the expression of some cytokines was slightly enhanced in the 0.9 LD50-exposed mice group (+183% ± 146% for G-CSF; +46% ± 26% for KC; +66% ± 35% for IL-1α). Interestingly, the expression of granulocyte-macrophage colony-stimulating factor (GM-CSF or CSF2) as well as IL-17 was slightly elevated in the 0.5 LD50-exposed mice group (+184% ± 131% and 86% ± 58%, respectively) (Figure 6).

### 3.4. Gut Modification

The consequences of chronic systemic low-level inflammation could have a significant impact on multiple physiological systems. Among the possible targets, we investigated the impact on intestinal function. A gut morphological evaluation was thus conducted 6 months post-NIMP exposure. Histological modifications were observed for both NIMP sublethal doses (Figure 7a). The quantification of goblet cell numbers and the ratio of intestinal villi size revealed a significant decrease in the 0.5 and 0.9 LD50-exposed mice groups (goblet cell numbers compared to CTL: 77.2% ± 8.3% and 78.7% ± 3.5%, respectively; intestinal villi size compared to CTL: 75.8% ± 3% and 69.6% ± 2.7%, respectively) (Figure 7b,c). Moreover, lymphoid foci numbers were significantly increased in both exposed mice groups (×3 ± 1.1 and ×5.5 ± 1.3 for 0.5 LD50 and 0.9 LD50, respectively) (Figure 7d). Finally, we conducted an evaluation of the Gram+/Gram− ratio. A dysbiosis of Gram− microbiota was observed in both NIMP-treated groups (Gram+/Gram− ratio: 0.55 ± 0.09 and 0.47 ± 0.11 for 0.5 LD50 and 0.9 LD50, respectively) (Figure 7e). This level of gut dysbiosis can trigger the innate immune response and chronic low-grade inflammation, leading to many age-related degenerative pathologies and unhealthy aging, which could in turn influence the gut microbiota composition [30].

### 3.5. Altered Microbiota Composition

A total of 788,900 16S rDNA reads was analyzed to establish the gut microbiota composition of the mice included in the present study. Our analyses led to the detection of 873 OTUs, evenly distributed among the mice groups (CTL: 551 ± 41, 0.5 LD50: 562 ± 54 and 0.9 LD50: 561 ± 44). No differences were detected regarding gut bacterial diversity and richness (Figure 8a) for mice challenged with NIMP 6 months post-exposure in comparison to the CTL group. As displayed in Figure 8b, nine predominant bacterial genera (Alistipes, Barnesiella, Bacteroides, Prevotella, Odoribacter, Alloprevotella, Clostridium_XIVa, Bilophila and Oscillibacter) shaped the gut microbiota composition of the mice, with highly similar abundances between groups. Furthermore, we observed a moderate shift in the gut microbiota composition of NIMP-exposed mice in comparison to the CTL group (Figure 8c). Two clusters consisting of CTL and NIMP-exposed mice (both 0.5 and 0.9 LD50 doses) were identified in a principal coordinate analysis (PCoA). This moderate shift can mostly be explained by higher abundances of Turicibacter (in both 0.5 and 0.9 LD50 groups) and Parabacteroides (0.9 LD50 group only) and a lower abundance of Coprococcus in NIMP-exposed mice compared to the CTL group (Figure 8d).

By combining other metadata related to anxiety-like behavior status and mononuclear blood cell percentages with gut microbiota composition at the genus level, we identified multiple significant associations linking the host with specific bacterial taxa (Figure 9). Notably, blood monocyte levels were positively correlated with the abundance of Aestuariispira (r = 0.62) and Parasutterella (r = 0.71) and negatively correlated with the levels of Bilophila (r = −0.61), Flavonifractor (r = −0.61), Gemmiger (r = −0.61), Hydrogenoanaerobacterium (r = −0.69), Oscillibacter (r = −0.67) and Pseudoflavonifractor (r = −0.70). Neutrophil and lymphocyte percentages were positively and negatively associated with Ruminococcus abundance (r = 0.75 and −0.77, respectively). The Intestinimonas genera was associated with anxiety status parameters such as cumulative open area activity (r = 0.62).

## 4. Discussion

The long-term follow-up of NA poisoning highlights the occurrence of neurological sequelae, even if the exposed victims only show transient low symptoms [2,31]. Psychological consequences are the main mood disorder reported by NA victims, and they can still be present more than 10 years after the incident [2,32]. Most of these complications affect emotions with increased fear, event recollections, irritability and agitation, which reflect anxiety-like behaviors. A growing concern in animal studies is the ability to evaluate behavioral and physiological modifications in order to better assess persistent long-term effects. We therefore developed an animal model of sublethal OP exposure that reproduces the early effects of sarin poisoning [17], which appears to be suitable for the study of the long-term effects of NA exposure.

In the present study, the use of NIMP as a sarin analog provided a robust replication of OP exposure symptoms and brain ChE activity inhibition. Similar parameters of intoxication severity have been observed in animals at early timepoints in a dose-dependent manner [17]. Mice subjected to the highest NIMP dose (0.9 LD50) showed marked brain ChE activity inhibition, behavioral alteration, weight loss and modification of WBC levels during the first week post-intoxication. Taken together, all of these signs are representative of the expected clinical features of OP poisoning and could be used to define a typical toxidrome of NA exposure consequences in animal subjects.

Animals exposed to 0.5 LD50 NIMP did not have any strong behavior or weight consequences, despite a significant inhibition of cerebral ChE activity and significant modification in WBC (lymphocytes, basophils and eosinophils) levels that persisted during the first week. One month after exposure, animals in both NIMP-exposed groups showed similar recovery, with no observed differences in weight or daily behavior (food intake, socialization, fur and body maintenance) compared to the CTL group. In fact, only cerebral ChE activity was still decreased in both groups. However, a significant increase in anxious behavior as evaluated by EPM occurred in both NIMP-exposed groups.

Anxiety disorder is one of the most reported disorders in animal models in response to OP exposure [10,33,34,35,36]. Generally, such behavioral assessments are conducted in animals exposed to high doses of OP that show persistent mood disorder defects [33,34]. Similar results have also been observed for the development of anxiety-like behavior in animals exposed to sublethal dose of OP [10,37] and are consistent with reports in humans [2,38,39]. Interestingly, in our study, significant anxiety disorder was observed 30 days after NIMP exposure in both groups (0.5 and 0.9 LD50), but was compensated for after the first month, and no significant anxious behaviors were seen before the fifth month post-exposure. This anxiety recurrence affected both NIMP-exposed groups and persisted up to 6 months post-intoxication.

It is worth noting that most of the anxiety tests used in rodents are based on a balance between a fearful response to an aversive condition (open, brightly lit or elevated spaces) and the tendency of animals to engage in exploratory activity or social interaction. To maintain the test novelty and the curiosity of the mice, and thus avoid any habituation, we chose to change the anxiety tests every month. Compensation of behavioral and/or cognitive deficits following OP exposure has already been reported in different rodent models [37,40]. Indeed, mice presented a partial improvement in cognitive performance over time in a Morris water maze and T-maze 3 months after soman exposure [40]. Transient improvement was also observed in rats 4 months after sarin vapor exposure, but no recovery was observed at longer timepoints [37]. Hence, our results are clearly in line with previous animal models, demonstrating that NIMP could be a reliable OP compound to reproduce NA exposure in the long-term.

The amygdala is the primary brain region involved in anxiety behavior [41,42] and is one of the main brain regions to be particularly affected by NA exposure [43,44,45]. Neuronal hyperexcitability has been noted in the basolateral amygdala following exposure to a high dose of soman due to a decrease in GABAergic inhibition in this area [35,36]. A slow recovery in AChE activity coupled to a loss of GABAergic neurons may explain this hyperactivity and the development of anxiety-like behavior [35,36]. However, the low doses of NIMP used in our study did not induce any observable neuropathologies at any timepoint studied. This result is consistent with our previous study [17] and with other animal models exposed to a low dose of NA [15,46]. Furthermore, no anatomical volume modification of the amygdala was observed in our model at any timepoint studied, suggesting that neither cell loss nor swelling was induced after the low-dose exposure to NIMP. The amygdala is particularly sensitive to stress, which could be reflected by an increase in inflammatory cytokines leading to its enlargement [41,47].

In accordance with the lack of any architectural change in the amygdala, no neuroinflammation process (i.e., significant elevated cytokine levels, astrocytes or microglial activation) was observed in either group of NIMP-exposed animals 6 months post-intoxication. Therefore, we suspected that peripheral inflammation could be involved in the establishment of anxiety-like behavior observed in NIMP-exposed animals in the longer term. Indeed, elevated peripheral inflammatory cytokine levels (IL−6, IL-1beta and TNF-alpha) have been associated with mood disorder development [41,47]. However, our evaluation of serum cytokines did not reveal any significant modifications in the inflammatory cytokines under study in the two groups 6 months post-NIMP exposure. On the other hand, a significant decrease in circulating basophil and eosinophil counts was observed 3 months after NIMP exposure and persisted for up to 6 months. A decrease in venous blood basophil counts has already been observed in patients with major depression disorder displaying elevated anxiety [48], suggesting that the alteration of leukocytes may play a role in the development of anxious behavior. Basophils and eosinophils are leukocyte subtypes particularly involved in allergic responses, but they are also involved in gut homeostasis regulation [49,50]. Since basophils and eosinophils play a role in the maintenance of the protective mucosal barrier and contribute to immune modulation towards gut microbes, these decreasing basophil and eosinophil counts may participate in the gut morphological alteration and dysbiosis observed in NIMP-exposed animals. We therefore decided to investigate if NIMP exposure could affect mice microbiota.

The gastrointestinal system expresses several nicotinic receptors and is highly innervated by the cholinergic neurons of the parasympathetic and enteric systems. In addition, ACh is involved in regulating several functions such as gut motility, local blood flow, intestinal mucosal barrier permeability and inflammation regulation [51,52,53]. Furthermore, it has been shown that ChE inhibition by soman, neostigmine, PB or DFP alters intestinal functions [54,55,56]. The intestine also plays a crucial role in the elimination of NA [57], which should impact the microbiota homeostasis. Indeed, one previous study found that the administration of PB along with the insecticide permethrin to mice alters the gut microbiome, with the enrichment of several bacterial families and genera in the treated animals [58].

Interestingly, the animal model used in the aforementioned study reproduced some of the GW symptoms reported by deployed veterans and showed abundant Coprococcus and Turicibacter correlated with neuroinflammation and gut leaching [58]. Moreover, a previous study conducted in humans reported that Coprococcus abundance is also associated with depressive disorder [59]. In our study, a decrease in Coprococcus and a concomitant increase in Turicibacter were observed in both groups 6 months after NIMP exposure, regardless of the dose received. The difference in Coprococcus variation could be explained by the exposure protocol and the long-term evaluation period, but an alternative possibility is that a lack of Coprococcus could be associated with anxiety.

To our knowledge, our study is the first demonstration of altered microbiota after NA exposure observed over the long term. In fact, several bacterial populations were found to be modified in rats 3 days after soman exposure, and no long-term change was reported (75 days post-exposure) where the gut microbiota remained resilient [60]. Although we assessed the gut microbiota composition at 6 months after NIMP exposure in order to determine if this could explain the second wave of anxiety-like behavior observed in NIMP exposed animals, we could not determine the timepoints at which these changes were implemented. Nevertheless, our microbiota analysis revealed a positive correlation between Intestinimonas abundance and the anxiety-like behavior level measured in mice exposed to NIMP. It is noteworthy that a positive link between Intestinimonas and psychological stress was previously reported in a rat model, in association with intestinal and blood-brain barrier alterations [61]. Furthermore, several bacterial species modified by NIMP exposure were correlated with the WBC variation observed in 0.5 LD50-exposed animals, which may enhance immune complications in these animals. Together, our results demonstrate that a single acute NA exposure can lead to long-term gut microbiota modifications.

## 5. Conclusions

In conclusion, our study demonstrates that exposure to a low dose of NA, identified as barely symptomatic for the 0.5 LD50 NIMP-exposed mice, could disrupt gut microbiota and immune homeostasis and alter emotional behavior in the long-term (Graphical abstract). Further studies are needed in order to decipher the mechanisms linking microbiota, anxiety-like behavior and immune cell response, as well as to discriminate more specific middle timepoints. The association between Intestinimonas and anxiety-like behavior requires further investigations to be confirmed. Nevertheless, our results highlight the need to care for all NA victims, even those exposed to low-doses, and to disclose new biomarkers that could be useful for follow-up with NA victims.

## Figures and Tables

**Figure 1 biomedicines-10-01167-f001:**
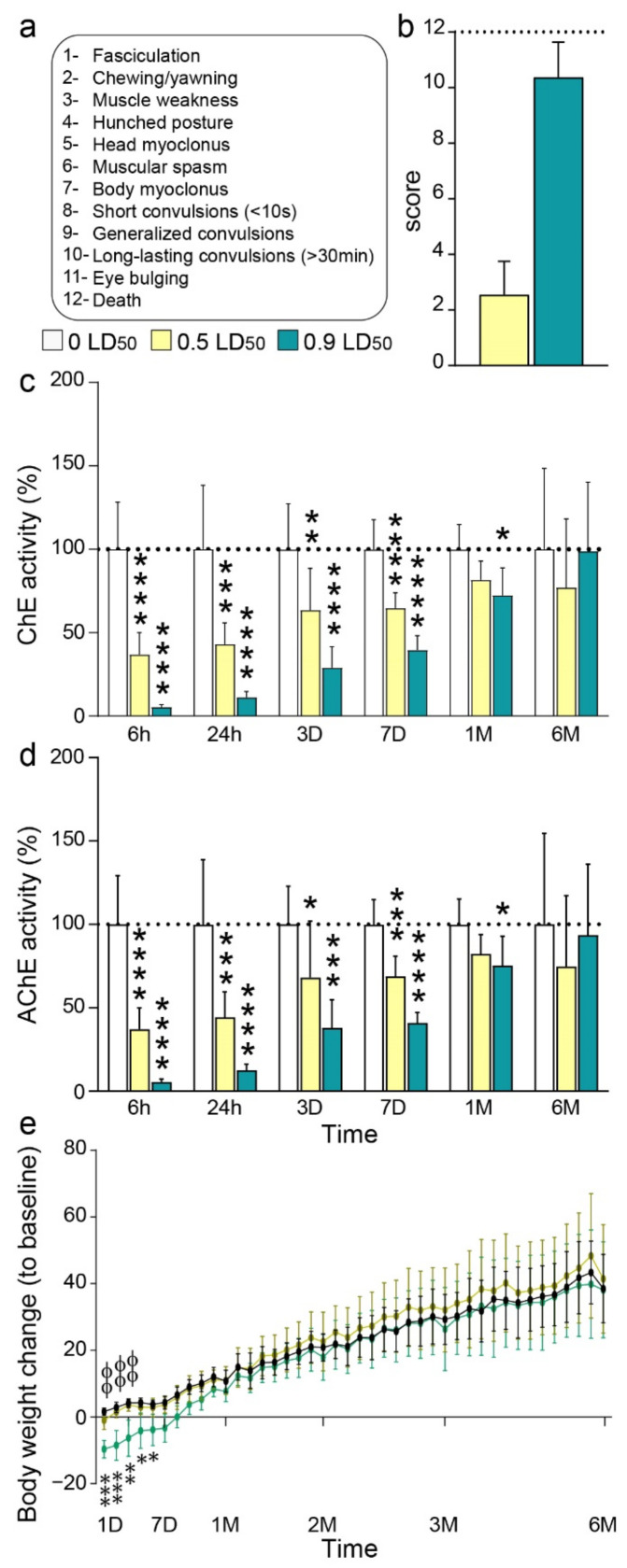
General effects of NIMP intoxication. (**a**) Intoxication scale based on the most relevant behavioral signs. (**b**) Average intoxication score induced by the two doses of interest: 2.55 ± 0.17 for 0.5 LD50 and 10.38 ± 0.16 for 0.9 LD50. (**c**) ChE activity at different timepoints post-NIMP exposure: 6 h (100 ± 11.5 for 0 LD50; 37.0 ± 4.6 for 0.5 LD50 and 5.4 ± 0.6 for 0.9 LD50); 24 h (100 ± 13.6 for 0 LD50; 43.0 ± 4.5 for 0.5 LD50 and 11.3 ± 1.3 for 0.9 LD50); 3 days (100 ± 9.6 for 0 LD50; 63.5 ± 8.8 for 0.5 LD50 and 29.0 ± 4.4 for 0.9 LD50); 7 days (100 ± 6.3 for 0 LD50; 64.8 ± 3.2 for 0.5 LD50 and 39.7 ± 3.3 for 0.9 LD50); 1 month (100 ± 6.7 for 0 LD50; 81.7 ± 5.0 for 0.5 LD50 and 72.5 ± 6.7 for 0.9 LD50) and 6 months (100 ± 17.2 for 0 LD50; 77.0 ± 13.7 for 0.5 LD50 and 98.9 ± 14.6 for 0.9 LD50); (*n* = 8 per group). Significant differences were determined by one-way ANOVA (F = 51.45, *p* < 0.0001 for 6H; F = 26.45, *p* < 0.0001 for 24H; F = 19.82, *p* < 0.0001 for 3D; F = 42.6, *p* < 0.0001 for 7D; F = 4.99, *p* = 0.02 for 1M and F = 0.76, *p* = 0.48 for 6M) with Dunnett’s post hoc test compared to the CTL group (**** *p* < 0.0001; *** *p* < 0.001; ** *p* < 0.01; * *p* < 0.05). (**d**) AChE activity at different timepoints post-NIMP exposure: 6 h (100 ± 11.9 for 0 LD50; 37.1 ± 4.5 for 0.5 LD50 and 5.3 ± 0.7 for 0.9 LD50); 24 h (100 ± 14.6 for 0 LD50; 44.3 ± 5.4 for 0.5 LD50 and 12.6 ± 1.3 for 0.9 LD50); 3 days (100 ± 8.7 for 0 LD50; 68.0 ± 12.0 for 0.5 LD50 and 37.9 ± 6.0 for 0.9 LD50); 7 days (100 ± 5.6 for 0 LD50; 68.9 ± 4.2 for 0.5 LD50 and 40.9 ± 2.3 for 0.9 LD50); 1 month (100 ± 6.8 for 0 LD50; 82.2 ± 5.2 for 0.5 LD50 and 75.3 ± 7.2 for 0.9 LD50) and 6 months (100 ± 19.3 for 0 LD50; 74.8 ± 14.1 for 0.5 LD50 and 93.5 ± 15.1 for 0.9 LD50); (*n* = 8 per group). Significant differences were determined by one-way ANOVA (F = 48.94, *p* < 0.0001 for 6H; F = 24.4, *p* < 0.0001 for 24H; F = 10.86, *p* = 0.0006 for 3D; F = 45.38, *p* < 0.0001 for 7D; F = 3.7, *p* = 0.04 for 1M and F = 0.67, *p* = 0.52 for 6M) with Dunnett’s post hoc test compared to the CTL group (**** *p* < 0.0001; *** *p* < 0.001; ** *p* < 0.01; * *p* < 0.05). (**e**) Body weight change relative to baseline (*n* = 14 for 0 LD50; *n* = 15 for 0.5 LD50 and *n* = 16 for 0.9 LD50 group). Significant differences were determined by two-way repeated measures ANOVA (F_Time_ = 282.8, *p* < 0.0001; F_Dose_ = 2.3, *p* = 0.11; F_TimexDose_ = 1.67, *p* = 0.0009) with Tukey’s multiple comparisons test (*** *p* < 0.001; ** *p* < 0.01; * *p* < 0.05 CTL vs. 0.9 LD50; ΦΦ *p* < 0.01 0.5 LD50 vs. 0.9 LD50).

**Figure 2 biomedicines-10-01167-f002:**
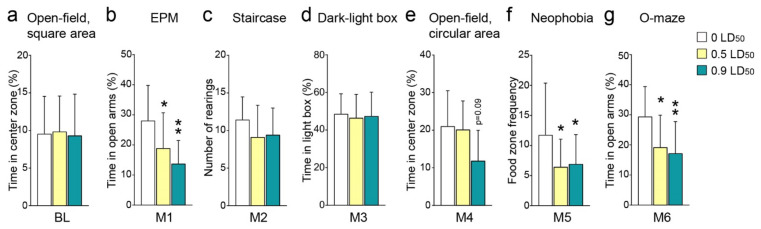
Development of anxiety-like behavior caused by NIMP intoxication. Anxiety-like behaviors were evaluated in NIMP-exposed and CTL animals using a different test every month (*n* = 14 for 0 LD50; *n* = 15 for 0.5 LD50 and *n* = 16 for 0.9 LD50 group). (**a**) Open-field, square area, at the baseline with the percentage of time spent in the center zone (9.6 ± 1.3 for 0 LD50; 9.9 ± 1.2 for 0.5 LD50 and 9.4 ± 1.4 for 0.9 LD50). (**b**) EPM at 1 month with the percentage of time spent in the open arms (28.1 ± 3.1 for 0 LD50; 18.9 ± 3.0 for 0.5 LD50 and 13.8 ± 1.9 for 0.9 LD50). (**c**) Staircase test, 2 months post-intoxication with the number of rearings on the top step (11.5 ± 0.8 for 0 LD50; 9.1 ± 1.1 for 0.5 LD50 and 9.4 ± 0.9 for 0.9 LD50). (**d**) Dark-light box at 3 months with the percentage of time spent in the light box (48.6 ± 2.8 for 0 LD50; 46.5 ± 3.2 for 0.5 LD50 and 47.5 ± 3.2 for 0.9 LD50). (**e**) Open-field, circular area, at 4 months with the percentage of time spent in the center zone (21.1 ± 3.8 for 0 LD50; 20.2 ± 2.7 for 0.5 LD50 and 11.9 ± 2.9 for 0.9 LD50). (**f**) Neophobia test at 5 months showing the frequency of visits to the center platform (11.8 ± 2.3 for 0 LD50; 6.5 ± 1.2 for 0.5 LD50 and 6.9 ± 1.2 for 0.9 LD50). (**g**) O-maze at 6 months post-exposure with the percentage of time spent in the open arms (29.4 ± 2.7 for 0 LD50; 19.2 ± 2.8 for 0.5 LD50 and 17.2 ± 2.6 for 0.9 LD50). Significant differences were determined by one-way ANOVA (F = 0.04, *p* = 0.9 for BL; F = 7.1, *p* = 0.002 for 1M; F = 1.6, *p* = 0.2 for 2M; F = 0.11, *p* = 0.89 for 3M; F = 2.8, *p* = 0.08 for 4M; F = 3.2, *p* = 0.04 for 5M and F = 5.4, *p* = 0.008 for 6M) with Dunnett’s post hoc test compared to the CTL group (** *p* < 0.01; * *p* < 0.05).

**Figure 3 biomedicines-10-01167-f003:**
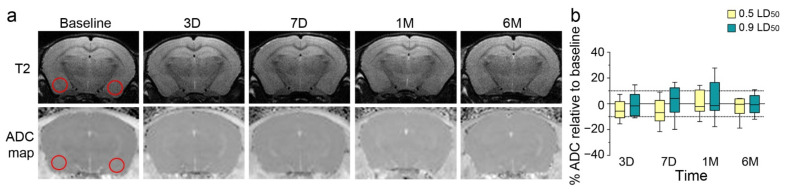
NIMP exposure does not induce morphological modifications or brain edema in the amygdala/piriform cortex. (**a**) T2 images with the apparent diffusion coefficient (ADC) map of one representative NIMP-exposed mouse (0.9 LD50) at different timepoints: before intoxication (baseline), and 3 days, 7 days, 1 month and 6 months after NIMP exposure. Regions of interest used for analyses are shown with red circles. (**b**) The percentage of ADC normalized to the baseline as a function of different timepoints for NIMP-exposed animal groups [(0.5 LD50: −5.1 ± 2.7 at 3D; −6.0 ± 3.5 at 7D; 1.7 ± 3.9 at 1M and −2.9 ± 3.1 at 6M) and (0.9 LD50: −0.8 ± 3.1 at 3D; 2.1 ± 4.3 at 7D; 2.9 ± 5.2 at 1M and −0.5 ± 2.7 at 6M)]. Statistical analyses were conducted by mixed-effects model (REML) analysis (F_Time_ = 0.89, *p* = 0.44; F_Dose_ = 1.5, *p* = 0.24; FT_imexDose_ = 0.36, *p* = 0.78) (*n* = 7 to 9 per group).

**Figure 4 biomedicines-10-01167-f004:**
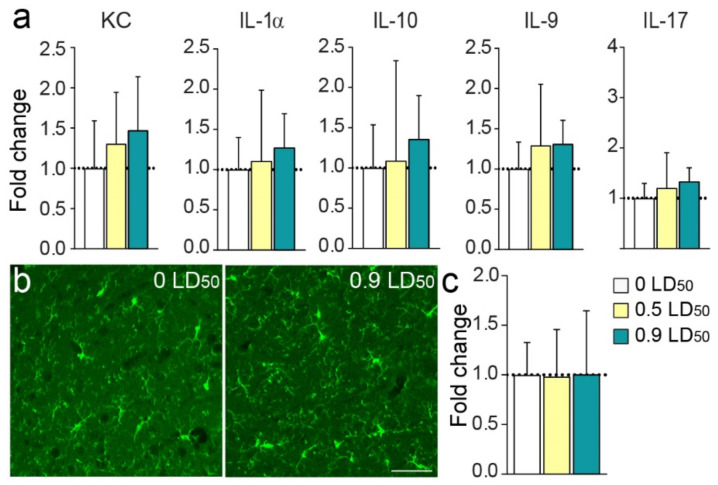
NIMP exposure does not induce long-term neuroinflammation in the amygdala/piriform cortex. (**a**) Expression levels of the cytokines KC, IL-1α, IL-10, IL-9 and IL-17 normalized to CTL values at 6 months post-intoxication [(KC: 1.0 ± 0.2 for 0 LD50; 1.3± 0.2 for 0.5 LD50 and 1.5 ± 0.2 for 0.9 LD50); (IL-1α: 1.0 ± 0.2 for 0 LD50; 1.1 ± 0.3 for 0.5 LD50 and 1.3 ± 0.2 for 0.9 LD50); (IL-10: 1.0 ± 0.2 for 0 LD50; 1.1 ± 0.5 for 0.5 LD50 and 1.4 ± 0.2 for 0.9 LD50); (IL-9: 1.0 ± 0.1 for 0 LD50; 1.3 ± 0.3 for 0.5 LD50 and 1.3 ± 0.1 for 0.9 LD50) and (IL-17: 1.0 ± 0.1 for 0 LD50; 1.2 ± 0.3 for 0.5 LD50 and 1.3 ± 0.1 for 0.9 LD50)]. Statistical analyses were conducted by one-way ANOVA (*n* = 8 per group). (**b**) IBA-1 labeling in the amygdala region of CTL (**left**) and 0.9 LD50 (**right**) mice. Scale bar = 50 µm. (**c**) Fold change of IBA1 labeling the optical density percentage normalized to CTL at 6 months post-intoxication: 1.0 ± 0.1 for 0 LD50; 0.98 ± 0.2 for 0.5 LD50 and 1.0 ± 0.2 for 0.9 LD50. Statistical analyses were conducted by Kruskal–Wallis test (*p* = 0.83; *n* = 7 per group).

**Figure 5 biomedicines-10-01167-f005:**
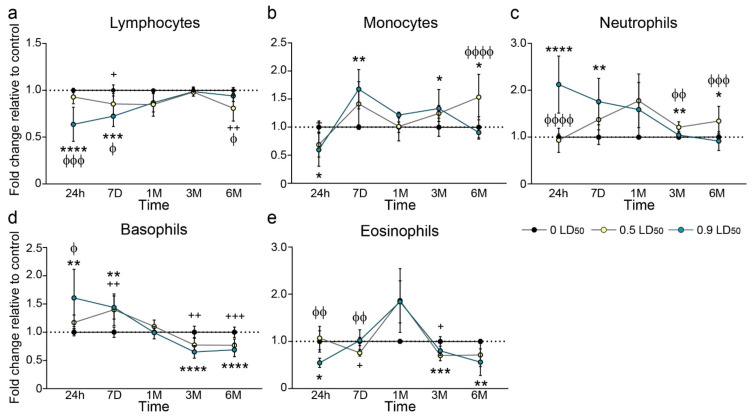
Evolution in white blood cell count after NIMP exposure. (**a**) Lymphocyte count at different timepoints after NIMP exposure relative to CTL: 24 h (1.0 ± 0.01 for 0 LD50; 0.93 ± 0.02 for 0.5 LD50 and 0.64 ± 0.06 for 0.9 LD50); 7 days (1.0 ± 0.02 for 0 LD50; 0.85 ± 0.03 for 0.5 LD50 and 0.72 ± 0.04 for 0.9 LD50); 1 month (1.0 ± 0.004 for 0 LD50; 0.85 ± 0.07 for 0.5 LD50 and 0.87 ± 0.05 for 0.9 LD50); 3 months (1.0 ± 0.003 for 0 LD50; 0.98 ± 0.01 for 0.5 LD50 and 0.99 ± 0.02 for 0.9 LD50) and 6 months (1.0 ± 0.12 for 0 LD50; 0.81 ± 0.05 for 0.5 LD50 and 0.94 ± 0.02 for 0.9 LD50); (*n* = 8 to 10 per group). Significant differences were determined by one-way ANOVA (F = 22.52, *p* < 0.0001 for 24H; F = 13.33, *p* = 0.0002 for 7D; F = 5.21, *p* = 0.02 for 1M; F = 0.32, *p* = 0.73 for 3M and F = 8.51, *p* = 0.0023 for 6M with Tukey’s multiple comparisons test. (**b**) Monocyte count at different timepoints after NIMP exposure relative to CTL: 24 h (1.0 ± 0.04 for 0 LD50; 0.69 ± 0.08 for 0.5 LD50 and 0.60 ± 0.13 for 0.9 LD50); 7 days (1.0 ± 0.02 for 0 LD50; 1.41 ± 0.12 for 0.5 LD50 and 1.68 ± 0.12 for 0.9 LD50); 1 month (1.0 ± 0.04 for 0 LD50; 1.01 ± 0.12 for 0.5 LD50 and 1.22 ± 0.02 for 0.9 LD50); 3 months (1.0 ± 0.06 for 0 LD50; 1.24 ± 0.05 for 0.5 LD50 and 1.33 ± 0.11 for 0.9 LD50) and 6 months (1.0 ± 0.07 for 0 LD50; 1.53 ± 0.13 for 0.5 LD50 and 0.90 ± 0.03 for 0.9 LD50); (*n* = 8 to 10 per group). Significant differences were determined by one-way ANOVA (F = 5.54, *p* = 0.015 for 24H; F = 6.39, *p* = 0.0065 for 7D; F = 2.88, *p* = 0.095 for 1M; F = 4.46, *p* = 0.024 for 3M and F = 15.78, *p* < 0.0001 for 6M with Tukey’s multiple comparisons test. (**c**) Neutrophil count at different timepoints after NIMP exposure relative to CTL: 24 h (1.0 ± 0.02 for 0 LD50; 0.93 ± 0.08 for 0.5 LD50 and 2.12 ± 0.23 for 0.9 LD50); 7 days (1.0 ± 0.06 for 0 LD50; 1.37 ± 0.11 for 0.5 LD50 and 1.76 ± 0.15 for 0.9 LD50); 1 month (1.0 ± 0.02 for 0 LD50; 1.77 ± 0.33 for 0.5 LD50 and 1.59 ± 0.29 for 0.9 LD50); 3 months (1.0 ± 0.01 for 0 LD50; 1.21 ± 0.04 for 0.5 LD50 and 1.04 ± 0.03 for 0.9 LD50) and 6 months (1.0 ± 0.03 for 0 LD50; 1.34 ± 0.11 for 0.5 LD50 and 0.92 ± 0.06 for 0.9 LD50); (*n* = 8 to 10 per group). Significant differences were determined by one-way ANOVA (F = 27.24, *p* < 0.0001 for 24H; F = 7.68, *p* = 0.0028 for 7D; F = 3.7, *p* = 0.067 for 1M; F = 12.19, *p* = 0.0003 for 3M and F = 9.50, *p* = 0.0009 for 6M with Tukey’s multiple comparisons test. (**d**) Basophil count at different timepoints after NIMP exposure relative to CTL: 24 h (1.0 ± 0.03 for 0 LD50; 1.17 ± 0.04 for 0.5 LD50 and 1.60 ± 0.17 for 0.9 LD50); 7 days (1.0 ± 0.03 for 0 LD50; 1.40 ± 0.09 for 0.5 LD50 and 1.44 ± 0.08 for 0.9 LD50); 1 month (1.0 ± 0.01 for 0 LD50; 1.10 ± 0.06 for 0.5 LD50 and 0.99 ± 0.04 for 0.9 LD50); 3 months (1.0 ± 0.05 for 0 LD50; 0.77 ± 0.04 for 0.5 LD50 and 0.65 ± 0.03 for 0.9 LD50) and 6 months (1.0 ± 0.03 for 0 LD50; 0.77 ± 0.03 for 0.5 LD50 and 0.69 ± 0.03 for 0.9 LD50); (*n* = 8 to 10 per group). Significant differences were determined by one-way ANOVA (F = 7.218, *p* = 0.0041 for 24H; F = 9.08, *p* = 0.0016 for 7D; F = 1.84, *p* = 0.19 for 1M; F = 14.12, *p* < 0.0001 for 3M and F = 20.04, *p* < 0.0001 for 6M with Tukey’s multiple comparisons test. (**e**) Eosinophil count at different timepoints after NIMP exposure relative to CTL: 24 h (1.0 ± 0.09 for 0 LD50; 1.07 ± 0.09 for 0.5 LD50 and 0.54 ± 0.05 for 0.9 LD50); 7 days (1.0 ± 0.04 for 0 LD50; 0.76 ± 0.02 for 0.5 LD50 and 1.02 ± 0.08 for 0.9 LD50); 1 month (1.0 ± 0.02 for 0 LD50; 1.87 ± 0.30 for 0.5 LD50 and 1.84 ± 0.26 for 0.9 LD50); 3 months (1.0 ± 0.04 for 0 LD50; 0.70 ± 0.05 for 0.5 LD50 and 0.80 ± 0.04 for 0.9 LD50) and 6 months (1.0 ± 0.02 for 0 LD50; 0.71 ± 0.08 for 0.5 LD50 and 0.56 ± 0.11 for 0.9 LD50); (*n* = 8 to 10 per group). Significant differences were determined by one-way ANOVA (F = 8.17, *p* = 0.0045 for 24H; F = 7.52, *p* = 0.0042 for 7D; F = 3.95, *p* = 0.058 for 1M; F = 12.92, *p* = 0.0007 for 3M and F = 6.13, *p* = 0.0088 for 6M) with Tukey’s multiple comparisons test. (**** *p* < 0.0001; *** *p* < 0.001; ** *p* < 0.01; * *p* < 0.05 CTL vs. 0.5 LD50; +++ *p* < 0.001; ++ *p* < 0.01; + *p* < 0.05 CTL vs. 0.9 LD50; ΦΦΦΦ *p* < 0.01; ΦΦΦ *p* < 0.001; ΦΦ *p* < 0.01; Φ *p* < 0.05 0.5 LD50 vs. 0.9 LD50).

**Figure 6 biomedicines-10-01167-f006:**
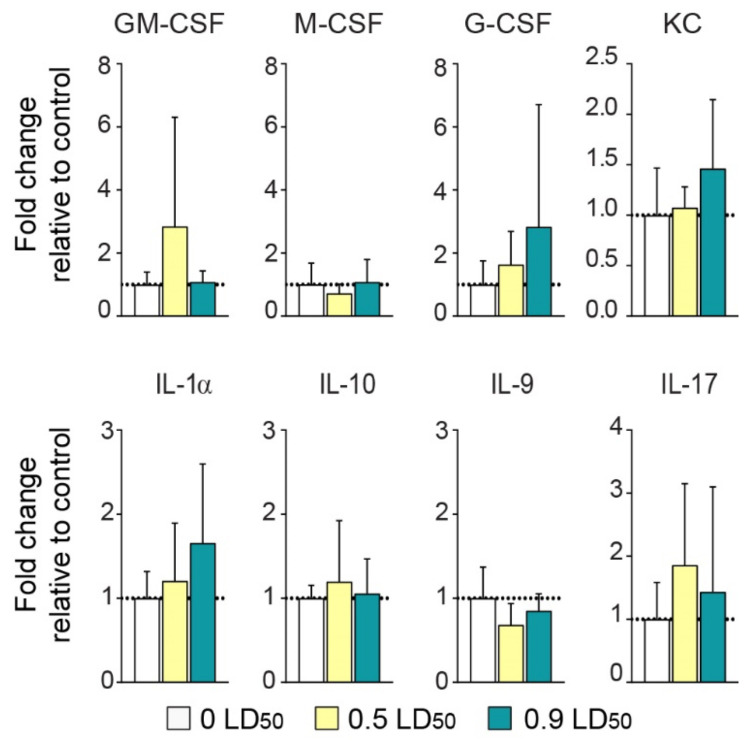
NIMP does not induce long-term systemic inflammation. Expression levels of the serum cytokines GM-CSF, M-CSF, G-CSF, KC, IL-1α, IL-10, IL-9 and IL-17 normalized to CTL values at 6 months post-intoxication [(GM-CSF: 1.0 ± 0.2 for 0 LD50; 2.8 ± 0.13 for 0.5 LD50 and 1.1 ± 0.01 for 0.9 LD50); (M-CSF: 1.0 ± 0.3 for 0 LD50; 0.72 ± 0.10 for 0.5 LD50 and 1.08 ± 0.3 for 0.9 LD50); (G-CSF: 1.0 ± 0.3 for 0 LD50; 1.63 ± 0.4 for 0.5 LD50 and 2.8 ± 1.46 for 0.9 LD50); (KC: 1.0 ± 0.2 for 0 LD50; 1.1± 0.08 for 0.5 LD50 and 1.46 ± 0.2 for 0.9 LD50); (IL-1α: 1.0 ± 0.1 for 0 LD50; 1.2 ± 0.3 for 0.5 LD50 and 1.7 ± 0.4 for 0.9 LD50); (IL-1α: 1.0 ± 0.07 for 0 LD50; 1.2 ± 0.3 for 0.5 LD50 and 1.06 ± 0.2 for 0.9 LD50); (IL-9: 1.0 ± 0.2 for 0 LD50; 0.68 ± 0.1 for 0.5 LD50 and 0.85 ± 0.07 for 0.9 LD50); (IL-17: 1.0 ± 0.3 for 0 LD50; 1.86 ± 0.6 for 0.5 LD50 and 1.43 ± 0.6 for 0.9 LD50)]. Statistical analyses were conducted by one-way ANOVA (*n* = 8 per group).

**Figure 7 biomedicines-10-01167-f007:**
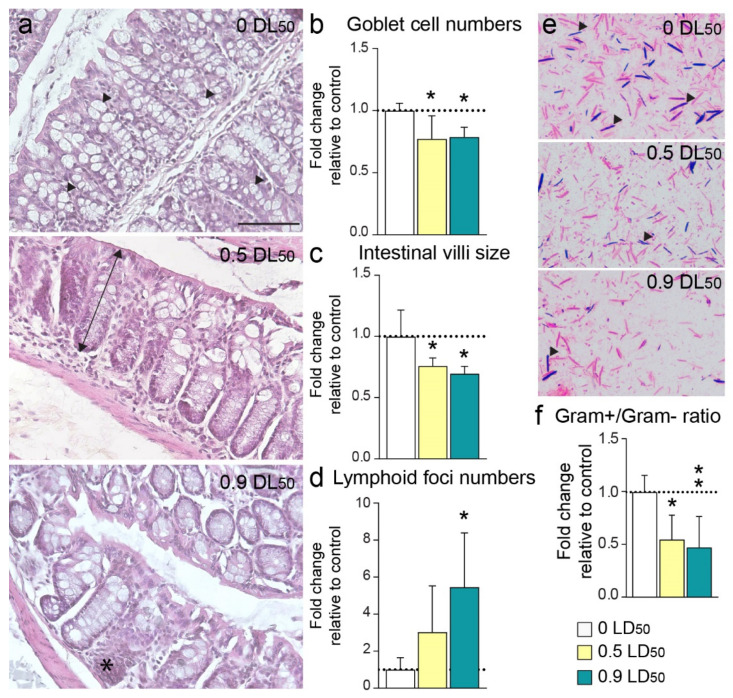
Long-term morphological changes in the large intestine induced by NIMP exposure. (**a**) Large intestine pictures of CTL (top panel), 0.5 LD50 (middle panel) and 0.9 LD50 (bottom panel) mice. Scale bar = 100 µm. Mucin-producing goblet cells (arrow heads) were identified in colon sections of CTL mice. The double arrow indicates an example of villus size in a 0.5 LD50 NIMP-exposed mouse; the asterisk indicates a lymphoid focus in a 0.9 LD50 NIMP-exposed mouse. Fold change in (**b**) goblet cell numbers (1.0 ± 0.03 for 0 LD50; 0.77 ± 0.08 for 0.5 LD50 and 0.79 ± 0.03 for 0.9 LD50), (**c**) villi size (1.0 ± 0.10 for 0 LD50; 0.76 ± 0.03 for 0.5 LD50 and 0.70 ± 0.03 for 0.9 LD50) and (**d**) lymphoid foci numbers (1.0 ± 0.28 for 0 LD50; 3.02 ± 1.12 for 0.5 LD50 and 5.46 ± 1.31 for 0.9 LD50) normalized to CTL values at 6 months post-intoxication. Significant differences were determined by one-way ANOVA (F = 5.5, *p* = 0.02 for goblet cell numbers; F = 6.87, *p* = 0.01 for villi size; F = 4.89, *p* = 0.028 for lymphoid foci numbers) with Tukey’s multiple comparisons test. (* *p* < 0.05 CTL vs. NIMP-treated mice). (**e**) Gram−stained stool smears in the large intestine of CTL (top panel), 0.5 LD50 (middle panel) and 0.9 LD50 (bottom panel) mice. Arrow heads illustrate Gram+ bacteria stained blue, while Gram− bacteria are stained pink (X100). (**f**) The ratio of Gram+/Gram− bacteria normalized to CTL animals (1.0 ± 0.7 for 0 LD50; 0.55 ± 0.09 for 0.5 LD50 and 0.47 ± 0.11 for 0.9 LD50). Significant differences were determined by one-way ANOVA (F = 7.6, *p* = 0.0047) with Tukey’s multiple comparisons test. (** *p* < 0.01; * *p* < 0.05 CTL vs. NIMP-treated mice).

**Figure 8 biomedicines-10-01167-f008:**
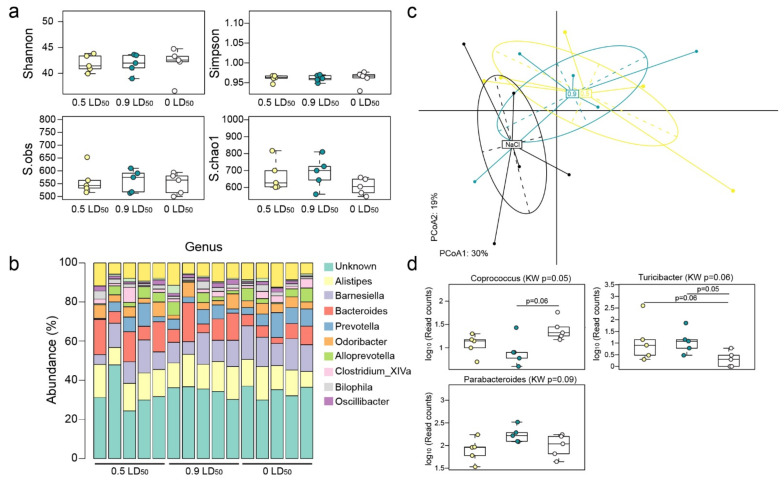
Mice exposed to NIMP harbor slight modifications in their gut microbiota composition 6 months after exposure. (**a**) Gut microbiota diversity was not affected by NIMP exposure. Microbial diversity and richness indexes were computed on the OTU abundance table. No difference was noted for these indexes between groups. (**b**) Overview of the gut microbiota composition at the genus level. Alistipes, Barnesiella, Bacteroides, Prevotella, Odoribacter, Alloprevotella, Clostridium_XIVa, Bilophila and Oscillibacter were the most abundant genera identified from mice gut microbiota. (**c**) NIMP exposure had a moderate impact on mice gut microbiota composition. Two-dimensional principal coordinates analysis was performed using the Bray–Curtis distances computed on the OTU abundance table. The total inertia explained (PCoA1 and PCoA2 axes) accounted for 49%. The CTL mice centroid (average microbiota profile) clustered away from both doses of NIMP-treated groups (0.5 LD50 in yellow and 0.9 LD50 in blue). This observation suggests that both doses of NIMP intoxication have a long-lasting effect on gut microbiota composition up to 6 months post-treatment. (**d**) Several gut bacterial genera display a shifted abundance following NIMP treatment. Differences in genus abundance between groups were assessed using the Kruskal–Wallis rank sum test (*p* ≤ 0.1) followed by a post hoc Dunn’s all-pairs rank comparison test. Three genera harbored a shifted abundance following NIMP exposure. In comparison to CTL mice, the 0.5 LD50 and 0.9 LD50 mice groups had a lower abundance of Coprococcus (*p* ≤ 0.05). The Turicibacter level increased in both 0.5 LD50 and 0.9 LD50 mice groups in comparison to the CTL group (*p* ≤ 0.06). Parabacteroides was slightly increased in the 0.9 LD50 group.

**Figure 9 biomedicines-10-01167-f009:**
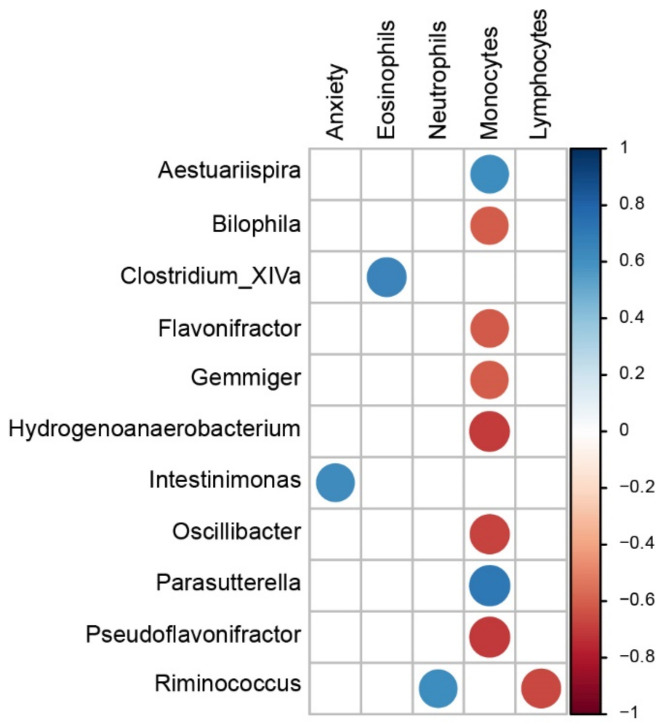
Pathological connection between modifications observed after NIMP exposure. Identification of bacterial genera found associated with host anxiety status and blood mononuclear cell abundance. Spearman correlations were computed using the genus abundance table and host metadata. Displayed correlations were all significant (*p* ≤ 0.05) and lower than −0.6 or higher than 0.6. Anxiety: cumulative open area activity (%), Eosinophils (%), Neutrophils (%), Monocytes (%) and Lymphocytes (%).

## Data Availability

Not applicable.

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
