# Peer review of "Long-Term Anxiety-like Behavior and Microbiota Changes Induced in Mice by Sublethal Doses of Acute Sarin Surrogate Exposure"

_biomedicines, 2022, doi:10.3390/biomedicines10051167_

Round 1

Reviewer 1 Report

In the manuscript entitled “Long-term anxiety and microbiota changes induced in mice by sublethal doses of acute sarin surrogate exposure”, François and colleagues investigated the long-term effect of a single low dose of the sarin surrogate 4-nitrophenyl isopropyl methylphosphonate (NIMP) in mice. NIMP sub-lethal exposure induced long-term anxiety and microbiota alterations.

The topic is interesting, the results are supported by a number of experiments, and the manuscript is well written.

However, some issues need to be addressed before publication.

Major concerns:

  • The authors correctly explained that the use of the sarin surrogate NIMP facilitated investigation in CWA-unauthorized laboratories, and that the molecule can reproduce several features of sarin intoxication. Is there any difference between sarin and NIMP in term of chemical/physical properties, pharmacokinetics, mechanism of action, and toxicity? If known, these differences should be clearly reported in the introduction section.
  • The authors reported that only male Swiss mice have been used in this study (line 97). Why only males and not females? The authors should explain this choice.
  • In relation to the above consideration, has any sexually dimorphic effect of sarin or NIMP been reported? Please, discuss this point.
  • The authors used a battery of tasks to evaluate anxiety in intoxicated mice. Correctly, to avoid any test habituation, a new anxiety test was performed every month. The results appear reliable, since the intoxicated mice displayed high anxiety-like behaviour in different tests. However, it is not clear how the authors selected the order of the behavioural tests. The reasons underlying the order of the tests should be explained in the “Materials and Methods” section.

Minor:

  • In the “Materials and Methods” section, some descriptions lack information [see for example the intoxication signs in the intoxication severity scale (line 115), Swiss-rolling technique (line 235), quantification of number of goblet cells and area of immune infiltrates through Histolab software (lines 239, 240)]. Please, be more detailed.
  • Describing the intoxication severity scale in the “Materials and Methods” section, the authors referred to 13 intoxication signs (line 115). In the “Results” section and in Figure 1a, the authors discussed about 12 signs (line 297). Please, be consistent.
  • Figure 7d: the line indicating the fold change of 1should be ahead the histogram.
  • Delete “6. Patents” section (page 668).

Reviewer 2 Report

In the manuscript “Long-term anxiety and microbiota changes induced in mice by sublethal doses of acute sarin surrogate exposure” the authors describe longitudinal changes in behavior, immunological measures, and gut microbiota changes”  The article is well written. Listed below are a few comments and questions.

Methods

Please clarify if the animals were group housed following exposure as well as prior to exposure.

Section 2.2 Should be “Mouse weight was” or “Mice weights were” in place of “Mice weight was..”

2.7 Statistics

Standard deviation (SD) is more appropriate instead of stand errors of means as SD will illustrate the variability within a group. Recommend editing graphs to show SD.

Results

3.1 Please include how many and what percentage of animals died in response to 0.9 LD50 NIMP.

Figure 1. It would be useful to see individual data points in the graph; at a minimum SD instead of SEM.

In Figure 2, the authors use the term “anxiety-like” behavior.  Since anxiety is a human trait, and it unclear whether an animal feels anxious, recommend using the term “anxiety-like” in other areas of the manuscript as well (title, introduction, discussion etc).

In addition, the tests used in this study are tests primarily developed to screen drugs for their ability to reduce anxiety-like behavior. These tests (light-dark test, open field, elevated plus maze) are based on the tendency of animal to explore novel environment verses the brightly lit open area and were developed to evaluate drugs that reduce “anxiety-like” responses (e.g. increase time in the open or bright area). Demonstrating an increase in anxiety-like behavior using these tests is challenging since control animals tend to spend most of their time in the perimeter of the open field, or the dark side of the box, or the enclosed arms. In the open field, the authors show that controls spend 10% time in the center zone, so decreasing time spent in the center is difficult. There may be “a floor effect” in several of these tests used, in that the animals already spend little time in the open making it less likely to see a reduction in time in the center.

Although the authors explain why they did not repeat tests out of concern of habituating to the test environment, the tests used are not all the same and the results obtained may relate to the test at that time point.  The elevated plus maze and zero maze are similar tests and have similar control time in open arms (30%) and showed a significant decrease in time in open arms in exposed animals at 1 month and 6 months respectively. Rather than a time course of change in behavior over time, the results may relate to the different tests used. One means to attempt to determine this would be to counterbalance the tests and have a gross measure of anxiety or to have separate groups at each time point.

Page 13, line 452, Figure 6. The authors indicate that IL-17 was elevated in the 0.5 LD50 exposed mice but does not show significance in the graph.  If significant, please add to the graph and the legend. If not significantly, the authors cannot say it was increased.  Also, indicate if standard deviation is shown.

Correlation coefficients between anxiety-like measures derived from unconditioned response tests with gut bacteria genera may be overly simplistic and should be interpreted with care. 

Reviewer 3 Report

In general, the topic of this paper is actual and important for the improvement of our knowledge of a complex mechanism of long term effects of sublethal poisoning of mice with nerve agents, the most important group of chemical warfare agents. The paper describes the clinical signs and symptoms, hippocampus ChE and AChE inhibition, anxious behavior, markers of systemic and intestinal inflammation and gut microbiota changes up to 6 months after subcutaneous administration of two  doses of sarin surrogate (0.5LD50 and 0.9LD50). The results clearly demonstrate that the administration of chosen doses of sarin surrogate caused mild or severe poisoning (according to the observation of clinical signs and symptoms and evaluation of cerebral AChE inhibition), long-term anxious behavior, intestinal inflammation and gut microbiota changes up to six-month post-exposure. On the other hand, anatomical changes and signs of neuroinflammation were not observed. Some results such as identification of long-term anxious behavior, systemic homeostasis disorganization and gut microbiota alteration following sarin surrogate sublethal exposure are original.

     The experimental methods used in this paper are modern and adequate to the investigated topic. The results are clearly demonstrated and relatively sufficiently discussed. However, the study suffers from some shortcomings (see below).

Shortcomings:

  • Materials and Methods - sarin surrogate was administered subcutaneously at two doses corresponding to 0.5LD50 and 0.9LD50. However the study, where the LD50 value of sarin surrogate for subcutaneous administration was evaluated, is not mentioned. In addition, why the low dose of sarin surrogate causing asymptomatic poisoning was  not used in this study? It would be interesting to know if asymptomatic poisoning is also connected with some long-term behavioral impairment and microbiota changes.
  • Material and Methods – why the alteration of cognitive functions such as memory and learning was not investigated besides the evaluation of anxious behavior? It is well known that long-term alteration of cognitive functions is characteristic for high-level as well as low-level exposure to nerve agents.
  • Material and methods – why the assessment of AChE activity was only done in hippocampus?
  • Results - very high inhibition of cerebral AChE was observed, especially after poisoning with the higher dose of sarin surrogate (up to 95% at 6 hours after poisoning). It is known that non-treated nerve agent-poisoned animals go very quickly to the death if the inhibition of AChE is higher than 90%. Therefore, is it necessary to describe how many poisoned mice survived after poisoning with both doses of sarin surrogate at mentioned time intervals, especially within 6 hours after poisoning. The information about the numbers of surviving and dead mice is missing.
  • Discussion - the recovery of ChE and AChE activity to the normal value at 6 months after poisoning is rather interesting because the synthesis of AChE de novo is very slow and no oxime treatment was used. In addition, it is not logic that the recovery of AChE activity is higher after higher dose of sarin surrogate. This fact should be explained in the study.
  • Discussion – the anxiety recurrence at 5 and 6 months after poisoning (compared to 3 and 4 months after poisoning) should be explained more thoroughly.
  • Conclusion – the first sentence should be changed because poisoning after lower dose of sarin surrogate was not asymptomatic.

Round 2

Reviewer 3 Report

The revised manuscript is improved compared to the prior version. All my reminders were sufficiently explained.